# Perceptions and experiences of the prevention, testing, and treatment of anaemia in pregnant women: A qualitative evidence synthesis

Ebony Verbunt[1]*, Martha Vazquez Corona[2], Özge Tunçalp[3], Lisa M. Rogers[4], Khic-Houy Prang[1], Cathy Vaughan[2], Meghan A. Bohren[2]

1 Centre for Health Policy, Melbourne School of Population and Global Health, The University of Melbourne, Melbourne, Victoria, Australia, 2 Gender and Women's Health Unit, Nossal Institute for Global Health, Melbourne School of Population and Global Health, The University of Melbourne, Melbourne, Victoria, Australia, 3 Department of Sexual and Reproductive Health and Research, World Health Organization (WHO), UNDP/UNFPA/UNICEF/WHO/World Bank Special Programme of Research, Development and Research Training in Human Reproduction, Geneva, Switzerland, 4 Department of Nutrition and Food Safety, World Health Organization (WHO), Geneva, Switzerland

* ebony.verbunt@unimelb.edu.au

## Abstract

Anaemia during pregnancy is a serious public health problem, inequitably burdening women in low-and middle-income countries. Despite numerous strategies and programs, anaemia prevalence rates have stagnated. We aimed to explore women's, health workers', and other key stakeholders' perceptions of anaemia in pregnant women, or their experiences with its prevention, testing, or treatment. We conducted a qualitative evidence synthesis. We searched MEDLINE (Ovid), Scopus, and CINAHL from inception to 12 November 2024, with no language or geographical restrictions. Data were analysed using thematic synthesis, and confidence in each review finding was assessed using the GRADE-CERQual approach. We included 61 papers from 23 countries. We grouped 25 review findings under four themes: (a) socio-cultural context of anaemia in pregnant women; (b) prevention and/or treatment of anaemia in pregnant women through diet, supplementation, or clinical intervention; (c) testing pregnant women for anaemia; and (d) factors affecting health workers' engagement in the management of anaemia in pregnant women. Women's management of anaemia in pregnancy was affected by the socio-cultural context, particularly their limited decision-making power and social position. Many women perceived a nutritious diet as integral to managing anaemia; however, high cost was often a barrier. Reasons women did not adhere to supplements included side-effects and difficulty remembering to take them, with family support improving adherence. Blood transfusion was perceived as treatment for severe anaemia, while intravenous iron was considered for women with iron-deficiency anaemia who were unable to take supplements, attended antenatal care late, or could not receive a transfusion. Health workers described difficulties testing pregnant women for anaemia and structural

**Data availability statement:** All relevant data are within the paper and its Supporting Information files.

**Funding:** EV is supported by the University of Melbourne Human Rights Scholarship. MAB's time is supported by an Australian National Health and Medical Research Council Investigator Grant (2025634) and Dame Kate Campbell Fellowship. The funders had no role in the study design, data collection and analysis, decision to publish, or preparation of the manuscript.

**Competing interests:** The authors have declared that no competing interests exist.

health-system deficiencies that affected their engagement. Findings may inform future WHO recommendations for managing anaemia in pregnant women. Future research could use review findings alongside implementation science frameworks to develop strategies for improving prevention, testing, and/or treatment of anaemia in pregnant women.

## Introduction

Anaemia during pregnancy is a serious public health problem, inequitably burdening women in low-and middle-income countries (LMICs) [1]. Women in LMICs are at a higher risk of anaemia compared to women in high-income countries due to greater likelihood of food insecurity, inadequate maternal nutrition, genetic haemoglobin disorders, infections such as malaria and HIV, and lower coverage of antenatal care [2,3]. In 2023, an estimated 29.2 million pregnant women worldwide had anaemia, including 10.9 million in Africa and 8.9 million in South-East Asia [4]. Effective management of anaemia during a woman's pregnancy requires implementation of prevention, testing, and treatment interventions [5]. The World Health Organization (WHO) recommends pregnant women take daily or intermittent oral iron and folic acid supplements to prevent and treat anaemia [5]. Depending on severity, underlying cause, and individual circumstances, women may be treated for anaemia through other modalities, including multiple micronutrient supplements, blood transfusion, or intravenous iron [6]. For example, in many high-income countries, guidelines recommend intravenous iron for women in the second or third trimester of pregnancy who have moderate to severe iron-deficiency anaemia and either do not respond to, or cannot tolerate, oral iron and folic acid supplements [7–9]. WHO also advise that women are tested for anaemia as part of routine antenatal care [5]. The recommended testing method is a full blood count using venous blood; however, in settings where this is impractical, a haemoglobinometer is recommended over a haemoglobin colour scale [5].

Anaemia in pregnant women is associated with a significant increase in mortality and morbidity [10]. For pregnant women, consequences of anaemia may include increased risk of infections, pre-eclampsia, and fatigue, leading to loss of work and ability to fulfill other responsibilities [11,12]. Anaemia during pregnancy also increases the consequences of serious birth complications such as postpartum haemorrhage [13]. Alarmingly, women who experience severe anaemia during pregnancy are twice as likely to die than pregnant women without severe anaemia [14]. Anaemia in pregnant women also has intergenerational consequences, with the baby at increased risk of pre-term birth, low birth weight, and perinatal mortality [12]. Babies that survive birth may endure impaired neurocognitive and motor development and are more likely to be anaemic themselves through childhood, adolescence, and their own pregnancy [6,15].

Despite numerous strategies and programs in place, anaemia prevalence rates have remained largely stagnant [4,16]. In 2012, the World Health Assembly (WHA)



adopted a 50% reduction of anaemia in women of reproductive age as a key Global Nutrition Target, which was added as an indicator for the United Nations Sustainable Development Goal 2 in 2020 [17]. However, when compared to other targets for nutrition, child and maternal health, progress in reducing anaemia is *'less than half the pace'* [16]. For example, the global prevalence of anaemia in pregnant women has reduced minimally, from an estimated 37.0% in 2012 to 35.5% in 2023 [4]. In 2023, WHO launched its first-ever comprehensive framework for action for accelerating anaemia reduction [16]. The framework acknowledges that the causes of anaemia extend beyond iron-deficiency, and that coverage and quality of interventions (e.g., oral iron and folic acid supplements) are low [16,18,19]. It calls for a multi-sectoral approach that builds on existing interventions to make progress toward the global target [16].

Qualitative evidence syntheses are increasingly used in decision-making processes to develop guidelines, such as WHO recommendations on antenatal care for a positive pregnancy experience [20]. Despite growing attention on anaemia, to our knowledge, there are no existing qualitative evidence syntheses about the perceptions and experiences of anaemia in pregnant women. A qualitative evidence synthesis focused specifically on anaemia management in pregnant women would provide valuable evidence to inform policy and practice, such as future updates to WHO antenatal care and anaemia management frameworks and guidelines. Therefore, we aimed to explore women's, health workers', and other key stakeholders' perceptions of anaemia in pregnant women, or experiences of preventing, testing, or treating anaemia in pregnant women.

## Methods

We followed the Cochrane Effective Practice and Organisation of Care (EPOC) qualitative evidence syntheses guidance [21] and the Preferred Reporting Items for Systematic Reviews and Meta-Analyses (PRISMA) guidelines (S1 Appendix). The protocol is registered with PROSPERO (CRD42023432537).

### Type of studies

We included primary qualitative and mixed-methods studies where the extraction of qualitative findings resulting from qualitative methods was possible. We included studies regardless of whether they were conducted alongside studies of the effectiveness of iron and folic acid supplementation during pregnancy [22,23]. No limitations were placed on publication date, language, or country. We excluded studies that used qualitative methods for data collection but did not perform a qualitative analysis. We excluded conference abstracts without a corresponding full paper.

### Topic of interest

The phenomenon of interest in this review was anaemia in pregnant women and interventions for the prevention, testing, and treatment of anaemia in pregnant women. We included studies that described the perceptions and experiences of multiple stakeholders: pregnant women, non-pregnant women, partners or other community members, skilled health workers, lay health workers, and other relevant stakeholders. We included studies that explored stakeholders' understanding of the causes of anaemia in pregnancy as well as studies that described all ways of preventing, testing, and treating anaemia in pregnant women. Studies conducted in any setting were included.

### Search methods for identification of studies

We searched MEDLINE (Ovid), Scopus, and CINAHL from inception to 12 November 2024. Search strategies were developed in consultation with an Information Specialist, and included a combination of terms relevant to anaemia, pregnant women, and qualitative research (S2 Appendix). We did not apply any limits on language, geographical setting, or publication date.

### Selection of studies

We imported title and abstracts into Covidence [24] and removed duplicates. Two review authors (EV and MVC) independently screened each study for potential inclusion, based on predefined criteria. Titles and abstracts of publications in

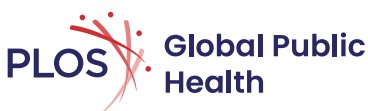

languages other than English were screened with the assistance of Google Translate. We retrieved the full text of all the studies identified as potentially relevant. Each full-text was assessed independently by two review authors (EV and MVC). One study published in Spanish that met the inclusion criteria was translated to English by a review author (MVC) who is a native Spanish speaker. Translation accuracy was then verified by another native Spanish speaker in our research network. We resolved disagreements by discussion, or when required, by involving a third review author (MAB).

### Data extraction and assessing methodological limitations

Data were extracted from the included studies using a form designed for this review. One review author (EV) extracted all data, and a second review author (MVC) checked all studies for accuracy and completion. The following information was extracted: study setting, sample characteristics, objectives, study design, data collection and analysis methods, qualitative themes, qualitative findings, supporting quotations, conclusions, and any relevant tables, figures or images.

Two review authors (EV and MVC) independently assessed the methodological limitations of the included studies using an adaptation of the Critical Skills Appraisal Programme (CASP) tool (www.casp-uk.net). Consensus was reached through discussion. We assessed aim, methodology, design, recruitment, data collection, data analysis, reflexivity, ethical considerations, and findings. We did not exclude studies based on our assessment of methodological limitations, but the results informed our assessment of confidence in the evidence.

### Data synthesis

We undertook a thematic synthesis to analyse data [25]. The first review author (EV) identified three studies that were highly relevant to the review question and used these studies to generate a free code list using NVivo 14 [26]. To develop a comprehensive codebook, one review author (EV) explored whether and how well the codes could be translated from one study to another by testing with another three particularly relevant studies; including new codes as necessary. Two review authors (EV and MVC) then independently read the remaining studies, reviewed coding outputs to identify similarities and differences between codes, then adjusted any new codes that emerged during the data analysis process. The first review author (EV) developed the initial thematic data outputs. These outputs were then consolidated into four overarching themes and 25 review findings through an iterative process of discussion with other review authors.

Two review authors (EV and MVC) used the GRADE-CERQual (Confidence in the Evidence from Reviews of Qualitative Research) approach [27,28] to assess confidence in each of the 25 review findings, based on the following four key components [29]:

1. Methodological limitations of included studies [30];

2. Coherence of the review finding [31];

3. Adequacy of the data contributing to a review finding [32]; and

4. Relevance of the included studies to the review question [33].

We assessed each component by level of concern – rated as no or very minor, minor, moderate, or serious. We then made a judgement about the overall confidence in the review finding – rated as high, moderate, low, or very low [29]. The final assessment was based on consensus among the review authors. We present GRADE-CERQual assessments in S3 Appendix.

### Review team reflexivity

We maintained a reflexive attitude throughout the stages of the review process, from study selection to data synthesis. The review author team represents diverse, international academic and professional backgrounds (social and behavioural sciences, implementation science, evaluation, public health, social epidemiology, medicine, nutrition) with a range of research focus areas and expertise. We were mindful that the perspectives of the review authors regarding subject

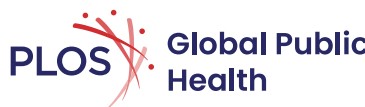

expertise, employment, perspectives of managing anaemia in pregnant women, and other background factors might influence our data collection, analysis, and interpretation process. For example, review authors with a medical background may interpret clinical intervention data more favourably than those with a nutrition background. To address this, we critiqued and challenged our preconceived assumptions through reflexive dialogues, and we supported each other to understand how these assumptions influenced the analysis or interpretation of findings. We believe that the diversity in our team helped us to critique and challenge our biases, and to develop review findings that were inclusive of and responsive to clinical practice. The team regularly discussed the progress of the review and critically explored all decisions, based on our collective and individual experiences (as women, clinicians, academics, and researchers).

## Results

Sixty-one papers from 60 studies met the inclusion criteria [34–94]. **Fig 1** presents the PRISMA flow diagram. **Table 1** presents the summarised characteristics of the included papers, while S4 Appendix provides more detailed individual characteristics for each included paper.

### Description of papers

Six papers were published between 2000–2012 [89–94] and 55 papers were published between 2013–2024 [34–88] (**Table 1**). Four papers were from studies conducted in multiple countries [34,76,83,93] and the remaining 57 papers were from studies conducted in a single country [35–75, 77–82,84–92,94].

The 61 papers present data from 23 countries and six geographic regions: 10 countries in Africa [34,37,39–41,45,47–49,53,55,58,60–62,64,68,69,72–76,78,79,83,84,87,91,93,94], six countries in South-East Asia [35,38,42–44,46,50–52,54,57,59,63,65–67,70,71,77,81,82,85,88,89,92,93], four countries in the Americas [90,93], one country in the Eastern Mediterranean [56,86,93], one country in the Western Pacific [80], and one country in Europe [36].

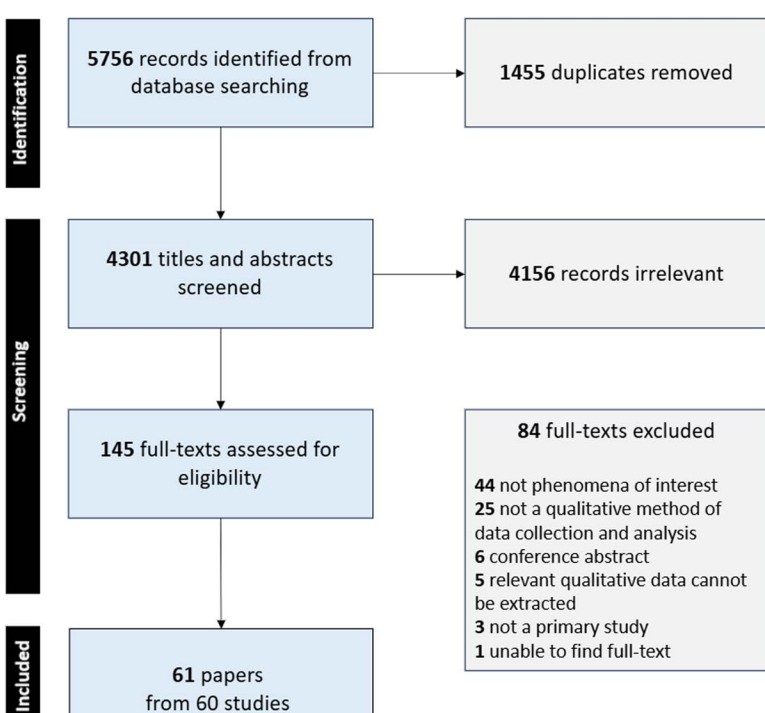

**Fig 1. PRISMA flow diagram depicting search and selection process.**

**Table 1. Summary of characteristics of included papers.**

| Characteristics | *n* =61 | % |
|---|---|---|
| **Year of study publication** | | |
| 2000–2012 | 6 | 9.8 |
| 2013–2024 | 55 | 90.2 |
| **Study region (*n* =64)[a]** | | |
| Africa | 31 | 48.4 |
| South-East Asia | 26 | 40.6 |
| Eastern Mediterranean | 3 | 4.7 |
| Americas | 2 | 3.1 |
| Western Pacific | 1 | 1.6 |
| Europe | 1 | 1.6 |
| **Resource level (*n* =64)[b]** | | |
| Low-income | 19 | 29.7 |
| Lower middle-income | 35 | 54.7 |
| Upper middle-income | 8 | 12.5 |
| High-income | 2 | 3.1 |
| **Type of participants (*n* =157)[c]** | | |
| Pregnant women | 40 | 25.5 |
| Non-pregnant women | 18 | 11.5 |
| Health workers | 44 | 28.0 |
| Lay health workers | 19 | 12.1 |
| Traditional birth attendants | 5 | 3.2 |
| Partner/family members | 16 | 10.2 |
| Cultural/religious/community leaders | 7 | 4.5 |
| Non-governmental organisation staff | 3 | 1.9 |
| Laboratory staff | 5 | 3.2 |

[a] One study conducted in Bolivia, Guatemala, Honduras (the Americas), Burkina Faso and Malawi (Africa), India and Indonesia (South-East Asia), and Pakistan (Eastern Mediterranean) [93].

[b] Three studies conducted in low-income and lower-middle income countries [34,76,93].

[c] Forty-three papers included more than one participant group [34–36,39,41,43–48,52,55–64,67–69,71–75,78,79,81,84–87,89–94].

Nineteen studies were conducted in a low-income country [34,41,48,49,58,63,68,72–74,76,83,84,87,89,91–94], 35 studies in a lower-middle income country [34,37,39,40,42–46,50–53,55–57,59–62,64,69–71,75–79,81,82,85,86,88,93], eight studies in an upper-middle income county [35,38,47,54,65–67,90], and two studies in a high-income country [36,80].

Most papers included the perspectives of multiple stakeholders. Forty papers included the perspectives of pregnant women [34–39,41–43,45,48,51–54,57,58,60–64,68–70,72–76,81,84–86,88–90,92–94], 18 papers included the perspectives of non-pregnant women [34,36,44,46,47,56,57,59,62,75,78,85–87,89–91,93], 44 papers included the perspectives of health workers [34,35,39–41,43–46,48,49,52,55–65,67–69,71–75,77–83,85,86,90,91,93,94], and 19 papers included the perspectives of lay health workers [44,45,47,52,55,58,59,64,67,71,73,75,81,84–87,91,93].

## Methodological limitations of included studies

The methodological limitations of the included studies are available in S5 Appendix. The main areas of concern were recruitment, reflexivity, ethics, and data analysis. It was rare for researchers to include a reflexivity statement, making it difficult to understand any power and positionality differences between researchers and participants. It was also

uncommon for researchers to detail or reference the method used for participant recruitment and data analysis. Some studies did not provide information on where they obtained ethics approval for their research.

### Confidence in the review findings

Of the 25 review findings, we used the GRADE-CERQual approach to grade one review finding as high confidence, 15 as moderate confidence, five as low confidence, and four as very low confidence. The evidence profile in S3 Appendix provides explanations for each GRADE-CERQual assessment. Our concerns about the confidence in the review findings were largely due to methodological limitations, relevance, and adequacy of the data. We had minor concerns (8 findings), moderate concerns (16 findings), and serious concerns (1 finding) related to methodological limitations, commonly because of no discussion of researcher reflexivity, inadequate explanation of participant recruitment, and incomplete information about data analysis. Most studies were conducted in the regions of Africa and South-East Asia (57/64) and in low-income and lower-middle income countries (54/64). This raised some concerns about relevance to other regions and upper-middle income and high-income countries. When assessing the adequacy of data (richness), we had some moderate concerns (5 findings) and serious concerns (7 findings) due to thin data supporting these review findings.

### Themes and findings from the qualitative evidence synthesis

We developed four overarching themes and 25 review findings (**Table 2**):

1.  Socio-cultural context of anaemia in pregnant women (six review findings);

2.  Prevention and/or treatment of anaemia in pregnant women through diet, supplementation, or clinical intervention (15 review findings);

3.  Testing pregnant women for anaemia (two review findings); and

4.  Factors affecting health workers' engagement in the management of anaemia in pregnant women (two review findings)

### Socio-cultural context of anaemia in pregnant women

Findings 1.1 to 1.6 are categorised under this theme, with 44 studies exploring women's and community members' knowledge and beliefs about anaemia in pregnant women, perceptions of traditional medicine, and the impact of women's social position and delayed or inconsistent antenatal care on anaemia management.

*Finding 1.1: Women and community members' knowledge of anaemia in pregnant women.* **Most women and community members were not aware of the term 'anaemia', and often referred to anaemia as 'lack of blood' or described it in terms of symptoms (e.g., fatigue and dizziness). Some women and community members had knowledge of more severe, long-term consequences of anaemia in pregnant women. Anaemia in pregnancy was often attributed to the woman sharing her blood with the fetus and an inadequate intake of nutritious food** (high confidence) [34,36,44–47,51,56–59,63,66,72–75,78,81,85,87–91,93,94].

Women and community members described how anaemia during pregnancy could lead to maternal death, stillbirth, premature birth, or stunted childhood development [34,56,58,72,73,88]. Other causes of anaemia in pregnant women outlined by participants included being pregnant at a young age, short spacing between pregnancies, stress, evil spirits, and working too much [44,57,78,91,93].

*Finding 1.2: Women and community members' beliefs about anaemia in pregnant women.* **Many women and community members considered common symptoms of anaemia (e.g., fatigue, weakness, and dizziness) as 'normal during pregnancy' and therefore not concerning. These beliefs were often passed down from older family members or from seeing other women in the community endure these symptoms during pregnancy. However, women sought care from health workers if anaemia symptoms hindered their ability to work, if they believed they would**

**Table 2. Summary of qualitative findings.**

| Findings | Summary of qualitative review finding | Contributing qualitative studies | Overall CERQual assessment | Explanation of overall assessment |
|---|---|---|---|---|
| | **1.0 Socio-cultural context of anaemia in pregnant women** | | | |
| 1.1 | **Women and community members' knowledge of anaemia in pregnant women.** Most women and community members were not aware of the term 'anaemia', and often referred to anaemia as 'lack of blood' or described it in terms of symptoms (e.g., fatigue and dizziness). Some women and community members had knowledge of more severe, long-term consequences of anaemia in pregnant women. Anaemia in pregnancy was often attributed to the woman sharing her blood with the fetus and an inadequate intake of nutritious food. | [34,36,44–47,51,56–59,63,66,72–75,78,81,85,87–91,93,94] | **High confidence** | No or very minor concerns about coherence, adequacy, and relevance; minor concerns about methodological limitations (recruitment, reflexivity, ethics, data collection, and data analysis). |
| 1.2 | **Women and community members' beliefs about anaemia in pregnant women.** Many women and community members considered common symptoms of anaemia (e.g., fatigue, weakness, and dizziness) as 'normal during pregnancy' and therefore not concerning. These beliefs were often passed down from older family members or from seeing other women in the community endure these symptoms during pregnancy. However, women sought care from health workers if anaemia symptoms hindered their ability to work, if they believed they would have a difficult childbirth, or if symptoms were perceived as affecting the fetus. | [35,50,51,54,56,57,59,75,78,79,88,90,93,94] | **Moderate confidence** | No or very minor concerns about adherence, minor concerns about adequacy (14 papers; 10 with moderate to thick data richness and 4 with thin data richness), minor concerns about relevance (4 regions; 2 upper-MICs, 8 lower-MICs, and 2 LICs), and moderate concerns about methodological limitations (qualitative methodology, research design, recruitment, reflexivity, ethics, data collection, data analysis). |
| 1.3 | **Positive perceptions about taking traditional medicine to manage anaemia in pregnancy.** Women, health workers, lay health workers, and community members described several reasons why women may take traditional medicines to prevent and/or treat anaemia in pregnancy. Reasons included where anaemia was attributed to black magic or a curse, beliefs that traditional medicine was more effective than Western medicine, challenges accessing antenatal care services, preference to not take medicine (i.e., pills) during pregnancy, and women having experienced side-effects with supplements. | [34,37,39,43,45,47,53,60,78,82,91,93] | **Low confidence** | No or very minor concerns about coherence, minor concerns about relevance (4 regions; 1 upper-MIC, 7 lower-MICs, and 7 LICs), moderate concerns about adequacy (12 papers; 3 with moderate to thick data richness and 9 with thin data richness), and moderate concerns about methodological limitations (recruitment, reflexivity, ethics, data collection, and data analysis). |
| 1.4 | **Fears about traditional medicine to manage anaemia in pregnancy.** Some women were pressured by family members (e.g., mothers-in-law) to take traditional medicine to prevent and/or treat anaemia in pregnancy. Women feared that if they did not take traditional medicine, it would be perceived as insubordination to family authority and they would be blamed for any complications during childbirth (e.g., caesarean section). Conversely, some women were concerned that taking traditional medicine may be harmful to the fetus and cause a miscarriage. | [53,75,91,94] | **Very low confidence** | No or very minor concerns about coherence, moderate concerns about methodological limitations (recruitment, reflexivity, ethics, data collection, and data analysis), serious concerns about adequacy (4 papers; 1 with thick data richness and 3 with thin data richness), and serious concerns about relevance (1 region; 2 lower-MICs and 2 LICs). |

*(Continued)*



**Table 2.** (Continued)

| Findings | Summary of qualitative review finding | Contributing qualitative studies | Overall CERQual assessment | Explanation of overall assessment |
|---|---|---|---|---|
| 1.5 | **Limited decision-making power and social position of women affects their management of anaemia in pregnancy.** Many women and some health workers and lay health workers reported that women were often not involved in decision-making around supplementation and diet during their pregnancy, with decisions made by the woman's family members (commonly their mother-in-law or husband). The heavy workload of women and social norms around prioritising family wellbeing over their own also limited women's ability to access anaemia management care. | [42,43,45,46,52,53,56,59,81,82,85,86,88,93] | Moderate confidence | No or very minor concerns about coherence, minor concerns about adequacy (14 papers; 8 with moderate to thick data richness and 6 with thin data richness), minor concerns about relevance (4 regions; 8 lower-MICs and 4 LICs), and moderate concerns about methodological limitations (recruitment, reflexivity, ethics, and data analysis). |
| 1.6 | **Delayed or inconsistent antenatal care affects the management of anaemia in pregnant women.** Many women, community members, health workers and lay health workers described how these delays affected the management of anaemia in pregnant women, such as insufficient time for women to take the recommended number of supplements. Women delayed or attended antenatal care inconsistently for various reasons, including a reluctance to disclose pregnancy due to fear of community gossip and vulnerability to evil spirits, limited healthcare decision-making power, distance to health facilities, long waiting times at health facilities, poor attitudes of health workers, the perception that care was unnecessary due to the absence of complications, and associated costs (e.g., transport). | [34,52,55–58,62,65,69,74,75,78,85,86,89,93] | Moderate confidence | No or very minor concerns about coherence, minor concerns about adequacy (16 papers; 10 with moderate to thick data richness and 6 with thin data richness), minor concerns about relevance (4 regions; 1 upper-MIC, 10 lower-MICs, and 7 LICs), and moderate concerns about methodological limitations (recruitment, reflexivity, ethics, and data analysis). |
| | **2.0 Prevention and/or treatment of anaemia in pregnant women through diet, supplementation, or clinical intervention** | | | |
| 2.1 | **Importance of a nutritious diet to manage anaemia in pregnancy.** Many health workers, lay health workers, women, and community members described that a nutritious diet was essential for preventing and/or treating anaemia in pregnant women, as they shared their food and blood with the fetus. Some women's diets changed during pregnancy to include more vegetables and fruits, larger portions, and to eat first rather than last within their families. Some women expressed a preference for managing anaemia through diet rather than supplementation, viewing food as natural and containing all necessary iron. | [34,36,43,44,46,52,56–58,66,70,73–75,78,79,81,85,87,88,90,91,93,94] | Moderate confidence | No or very minor concerns about coherence and relevance, minor concerns about adequacy (24 papers; 12 with moderate to thick data richness and 12 with thin data richness), and moderate concerns about methodological limitations (recruitment, reflexivity, ethics, data collection, and data analysis). |
| 2.2 | **Challenges to women having a nutritious diet to manage anaemia in pregnancy.** Many women and some community members, health workers, and lay health workers described how the high cost of nutritious food was a barrier to women having a nutrient-rich diet to prevent and/or treat anaemia in pregnancy. Additional barriers to women having a nutritious diet in pregnancy included limited decision-making power and social positionality of women, low availability of nutritious foods in some settings, inadequate knowledge of iron-rich foods, consumption of food and drink that inhibited iron absorption, and food taboos and religious beliefs that prevented pregnant woman from eating certain foods (e.g., animal products). | [35,42,44,46,50–54,56,57,65,70,73–75,77,81,82,85,88,93,94] | Moderate confidence | No or very minor concerns about coherence and relevance, minor concerns about adequacy (23 papers; 16 with moderate to thick data richness and 7 with thin data richness), and moderate concerns about methodological limitations (recruitment, reflexivity, ethics, data collection, and data analysis). |

*(Continued)*

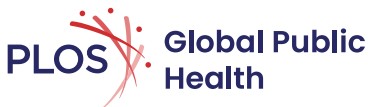

Table 2. (Continued)

| Findings | Summary of qualitative review finding | Contributing qualitative studies | Overall CERQual assessment | Explanation of overall assessment |
|---|---|---|---|---|
| 2.3 | **Characteristics of supplements.** Some women discussed the bitter taste, smell, and large size of supplements as negatively affecting adherence. | [35,38,39,42,43,47,52,59,60,63,75] | **Low confidence** | No or very minor concerns about coherence, minor concerns about methodological limitations (recruitment, reflexivity, data collection, and data analysis), serious concerns about adequacy (11 papers; 1 with moderate to thick data richness and 10 with thin data richness), and moderate concerns about relevance (2 regions; 2 upper-MIC and 3 lower-MICs). |
| 2.4 | **Side-effects of supplements.** Many women reported not adhering to supplements after experiencing side-effects, including nausea, dizziness, vomiting, diarrhoea, constipation, blackened stool, heartburn, loss or increase of appetite, and/or indigestion. | [34–39,41–43,45–47,52, 53,57–60,62,63,72–75, 84–86,88,93] | **Moderate confidence** | No or very minor concerns about coherence and relevance, moderate concerns about adequacy (29 papers; 6 with moderate to thick data richness and 23 with thin data richness), and moderate concerns about methodological limitation (recruitment, reflexivity, ethics, data collection, and data analysis). |
| 2.5 | **Misconceptions about supplements.** Women, community members, and some health workers and lay health workers had misconceptions about supplements that limited women from taking them to prevent and/or treat anaemia in pregnancy. Common misconceptions included that supplements would increase the size of the fetus and result in women having a difficult childbirth, were only needed to treat women with anaemia, could cause miscarriage, were bad for the fetus or woman's health, or increased maternal blood and therefore bleeding during birth. These misconceptions were often told to women by family members (e.g., mother-in-law). | [35,37,43,46,47,50,52, 57,59,60,62,63,65,85, 86,90,92–94] | **Moderate confidence** | No or very minor concerns about coherence, minor concerns about relevance (4 regions; 3 upper-MICs, 9 lower-MICs, and 3 LICs), moderate concerns about adequacy (19 papers; 8 with moderate to thick data richness and 11 with thin data richness), and serious concerns about methodological limitations (recruitment, reflexivity, ethics, data collection, and data analysis). |
| 2.6 | **Challenges to women remembering to take supplements**. Many women forgot to take supplements consistently, often due to competing daily activities, such as household chores and work, caring for children, being away from home, or taking other medications. | [36,38,39,41,43,47,53, 54,57,59,62,63,68,72–75, 84,86,92,93] | **Moderate confidence** | No or very minor concerns about coherence and relevance, moderate concerns about adequacy (21 papers; 5 with moderate to thick data richness and 16 with thin data richness), and moderate concerns about methodological limitations (recruitment, reflexivity, ethics, and data analysis). |
| 2.7 | **Inadequate counselling from health workers and lay health workers about supplements.** Women received insufficient and/or inconsistent information on – why they were provided with the supplements, the benefits of taking supplements, when they should start taking supplements, how often they should take supplements, potential side-effects of supplements, and how side-effects were minimised. | [35,38,43,47,54,57,58,60, 62,73–75,86,89,90,93] | **Moderate confidence** | No or very minor concerns about coherence, minor concerns about adequacy (16 papers; 11 with moderate to thick data richness and 5 with thin data richness), minor concerns about relevance (4 regions, 3 upper-MICs, 7 lower-MICs, and 6 LICs), and moderate concerns about methodological limitations (recruitment, reflexivity, ethics, data collection, and data analysis). |

*(Continued)*

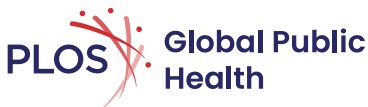

**Table 2.** (Continued)

| Findings | Summary of qualitative review finding | Contributing qualitative studies | Overall CERQual assessment | Explanation of overall assessment |
|---|---|---|---|---|
| 2.8 | **Insufficient supply of supplements.** Many health workers, lay health workers, and some women described an insufficient supply of supplements, which resulted in health workers and lay health workers prioritising the distribution of supplements to anaemic pregnant women over non-anaemic pregnant women, providing women with an inadequate quantity of supplements, or referring women to other facilities to purchase supplements out-of-pocket (e.g., private drug stores). | [35,41,45,46,49,55,59,60,62,64,69, 71,73,75,77,85,86,91,93] | **Moderate confidence** | No or very minor concerns about coherence, minor concerns about adequacy (19 papers; 10 with moderate to thick data richness and 9 with thin data richness), minor concerns about relevance (4 regions; 10 lower-MICs and 4 LICs), and moderate concerns about methodological limitations (recruitment, reflexivity, ethics, data collection, and data analysis). |
| 2.9 | **Women's knowledge of supplements and experience of positive benefits.** Adherence to supplements was motivated by women's knowledge of their health benefits, particularly for the fetus, and for the prevention of illness and excessive blood loss during childbirth. Some women discussed how the relief of anaemia symptoms after taking supplements (e.g., no longer feeling weak or lightheaded) improved adherence. | [36,38,41,47,52,53,60,62,68,73–75,85,86,88,90–94] | **Moderate confidence** | No or very minor concerns about coherence and relevance, minor concerns about adequacy (20 papers; 11 with moderate to thick data richness and 9 with thin data richness), and moderate concerns about methodological limitations (recruitment, reflexivity, ethics, and data analysis). |
| 2.10 | **Influence of family support on women taking supplements.** Many women, health workers, lay health workers, and community members highlighted how women's adherence to supplements was influenced by whether they had received encouragement and/or reminders to take supplements from family members (most commonly their husband, mother-in-law, or parents). | [38,39,43,47,50–52, 57,63,66,68,73, 74,76,82,86,92,94] | **Moderate confidence** | No or very minor concerns about coherence, minor concerns about adequacy (18 papers; 11 with moderate to thick data richness and 7 with thin data richness), minor concerns about relevance (3 regions; 1 upper-MIC, 5 lower-MICs, and 3 LICs), and moderate concerns about methodological limitations (recruitment, reflexivity, ethics, and data analysis). |
| 2.11 | **Reminder strategies to take supplements.** Some women and health workers perceived reminder strategies as facilitating women's adherence to supplements. Reminder strategies included linking taking supplements with other daily behaviour (e.g., with mealtime) and storing supplements in a visible location. | [36,57,73,75] | **Very low confidence** | No or very minor concerns about coherence, minor concerns about methodological limitations (recruitment, reflexivity, and data analysis), serious concerns about adequacy (4 papers; 1 with moderate data richness and 3 with thin data richness), and serious concerns about relevance (3 regions; 1 HIC, 2 lower-MICs and 1 LIC). |
| 2.12 | **Recommendation from health workers to take supplements.** Many women reported taking supplements because they trusted and accepted the advice of health workers. | [38,41,52,57,59,60,62, 63,68,73,74,85–88,94] | **Moderate confidence** | No or very minor concerns about coherence, minor concerns about adequacy (16 papers; 6 with moderate to thick data richness and 10 with thin data richness), minor concerns about relevance (3 regions; 1 upper-MIC, 6 lower-MICs and 4 LICs), and moderate concerns about methodological limitations (recruitment, reflexivity, ethics, and data analysis). |

*(Continued)*

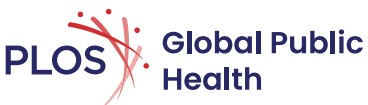

**Table 2.** (Continued)

| Findings | Summary of qualitative review finding | Contributing qualitative studies | Overall CERQual assessment | Explanation of overall assessment |
|---|---|---|---|---|
| 2.13 | **Alternative distribution methods for supplements.** Most women, health workers, and lay health workers supported distributing supplements to women during home visits. Lay health workers discussed how women's adherence to supplements improved with home visits, as they also allowed for health education and counselling. Some women discussed how community-based distribution mitigated barriers to receiving supplements, such as distance to health facilities and transportation costs. | [47,64,73,89] | **Very low confidence** | No or very minor concerns about coherence, minor concerns about methodological limitations (reflexivity and data analysis), serious concerns about adequacy (4 papers; 3 with moderate to thick data richness and 1 with thin data richness), and serious concerns about relevance (2 regions; 1 upper-MIC, 1 lower-MIC, and 2 LICs). |
| 2.14 | **Blood transfusion as treatment for severe anaemia in pregnant women.** Women and health workers perceived blood transfusion as treatment for women with severe anaemia; however, cost, religion, blood and donor availability, fear of contracting diseases, and a woman's social position limited access. | [34,39,45,52,54,57,80,87,93] | **Low confidence** | No or very minor concerns about coherence, moderate concerns about methodological limitations (reflexivity, ethics, and data analysis), moderate concerns about relevance (3 regions; 1 HIC, 1 upper-MIC, 4 lower-MICs, and 3 LICs), and serious concerns about adequacy (9 papers; 2 with moderate to thick data richness and 7 with thin data richness). |
| 2.15 | **Intravenous iron as an alternative treatment option for iron-deficiency anaemia in pregnant women.** Health workers, women, and community members described intravenous iron as a treatment option for women who had iron-deficiency anaemia and were unable to take supplements (e.g., due to side-effects), who attended antenatal care late, or who were unable to have a blood transfusion because of religious beliefs. | [39,40,48,56,57,80,90] | **Low confidence** | No or very minor concerns about coherence, moderate concerns about methodological limitations (recruitment, reflexivity, ethics, data collection, and data analysis), moderate concerns about relevance (5 regions; 1 HIC, 1 upper-MIC, 3 lower-MICs, and 1 LIC), and serious concerns about adequacy (7 papers; 4 with thick data richness and 3 with thin data richness). |
| | **3.0 Testing pregnant women for anaemia** | | | |
| 3.1 | **Benefits of testing pregnant women for anaemia with point-of-care devices using capillary blood.** Health workers provided with anaemia point-of-care devices (portable haemoglobinometers or a colour scale) felt they were easy to use with capillary blood and were eager to meet pregnant women's antenatal care needs. Health workers described how more women were diagnosed with anaemia as they were no longer lost to follow-up during the referral process to laboratories for a full blood count using venous blood. Women described how haemoglobin results from a point-of-care test improved their trust in the anaemia diagnosis and increased their involvement in the treatment decision-making process. | [61,69] | **Very low confidence** | No or very minor concerns about coherence, minor concerns about methodological limitations (reflexivity), serious concerns about adequacy (2 papers; 2 with thick data richness), and serious concerns about relevance (1 region; 2 lower-MICs). |

*(Continued)*

**Table 2.** (Continued)

| Findings | Summary of qualitative review finding | Contributing qualitative studies | Overall CERQual assessment | Explanation of overall assessment |
|---|---|---|---|---|
| 3.2 | **Challenges to testing pregnant women for anaemia with point-of-care devices using capillary blood and full blood count using venous blood.** Challenges to health workers using anaemia point-of-care devices with capillary blood included a shortage of tests, staff, and equipment (e.g., cuvettes required for most haemoglobinometers), inadequate training, and heavy workloads. For full blood count using venous blood, challenges described by health workers included a lack of laboratory facilities and women not attending facilities when referred. Health workers expressed reservations about the accuracy of results for both types of anaemia tests. | [34,45,46,61,65,67,69,75,77,82,83] | **Low confidence** | No or very minor concerns about coherence, minor concerns about methodological limitations (recruitment and reflexivity), minor concerns about relevance (2 regions; 1 upper-MIC, 5 lower-MICs, and 1 LIC), and moderate concerns about adequacy (11 papers; 7 with thick data richness and 4 with thin data richness). |
| | **4.0 Factors affecting health workers' engagement in the management of anaemia in pregnant women** | | | |
| 4.1 | **Training and guidance on how to manage anaemia in pregnant women among health workers and lay health workers.** Some health workers and lay health workers described receiving inadequate training on how to manage anaemia in pregnant women (e.g., on counselling, supplements, and/or testing). They also described a lack of specific or standardised guidelines on how to manage anaemia in pregnant women. | [34,45,49,61,62,64,65,67,69,71,73, 75,77,80,82,85] | **Moderate confidence** | No or very minor concerns about coherence, minor concerns about methodological limitations (recruitment, reflexivity, and data analysis), minor concerns about adequacy (16 papers; 8 with moderate to thick data richness and 8 with thin data richness), and minor concerns about relevance (3 regions; 1 HIC, 1 upper-MIC, 6 lower-MICs, and 2 LICs). |
| 4.2 | **Health worker and lay health worker challenges to managing anaemia in pregnant women.** Many health workers and lay health workers described how staffing shortages, heavy workloads, insufficient facilities and space, and poor work conditions affected their ability to manage anaemia in pregnant women. | [34,45,55,57,60,62,65,67,69,71,73 ,77,82,83] | **Moderate confidence** | No or very minor concerns about coherence, minor concerns about methodological limitations (recruitment, reflexivity, and data analysis), minor concerns about adequacy (14 papers; 7 with moderate to thick data richness and 7 with thin data richness), and minor concerns about relevance (2 regions; 1 upper-MIC, 5 lower-MICs, and 2 LICs). |

**have a difficult childbirth, or if symptoms were perceived as affecting the fetus** (moderate confidence) [35,50,51,54, 56,57,59,75,78,79,88,90,93,94].

In several countries, women and community members described feeling fatigued and weak as a normal part of pregnancy [35,50,51,54,56,75,88,90,93], *"just like a woman puts on weight"* [88].

*Finding 1.3: Positive perceptions about taking traditional medicine to manage anaemia in pregnancy.* **Women, health workers, lay health workers, and community members described several reasons why women may take traditional medicines to prevent and/or treat anaemia in pregnancy. Reasons included where anaemia was attributed to black magic or a curse, beliefs that traditional medicine was more effective than Western medicine, challenges accessing antenatal care services, preference to not take medicine (i.e., pills) during pregnancy, and women having experienced side-effects with supplements** (low confidence) [34,37,39,43,45,47,53,60,78,82,91,93].

Traditional medicine was discussed in general terms rather than specifically, often referred to as plants, herbs, tonics, or concoctions [34,37,39,45,60,78,93].

*Finding 1.4: Fears about traditional medicine to manage anaemia in pregnancy.* **Some women were pressured by family members (e.g., mothers-in-law) to take traditional medicine to prevent and/or treat anaemia in pregnancy. Women feared that if they did not take traditional medicine, it would be perceived as insubordination to family authority and they would be blamed for any complications during childbirth (e.g., caesarean section). Conversely, some women were concerned that taking traditional medicine may be harmful to the fetus and cause a miscarriage** (very low confidence) [53,75,91,94].

Culture was a powerful reason why women chose traditional medicine over supplements to manage anaemia during pregnancy [53]. In Nigeria, the government retrained herbalists and traditional practitioners to ensure the safety of their practices [94].

*Finding 1.5: Limited decision-making power and social position of women affects their management of anaemia in pregnancy.* **Many women and some health workers and lay health workers reported that women were often not involved in decision-making around supplementation and diet during their pregnancy, with decisions made by the woman's family members (commonly their mother-in-law or husband). The heavy workload of women and social norms around prioritising family wellbeing over their own also limited women's ability to access anaemia management care** (moderate confidence) [42,43,45,46,52,53,56,59,81,82,85,86,88,93].

Many women described the expectation to prioritise their families' well-being over their own; some even felt the need to move back to their family homes during pregnancy to receive better care than they would have at their in-laws' [52,59].

*Finding 1.6: Delayed or inconsistent antenatal care affects the management of anaemia in pregnant women.* **Many women, community members, health workers and lay health workers described how these delays affected the management of anaemia in pregnant women, such as insufficient time for women to take the recommended number of supplements. Women delayed or attended antenatal care inconsistently for various reasons, including a reluctance to disclose pregnancy due to fear of community gossip and vulnerability to evil spirits, limited health-care decision-making power, distance to health facilities, long waiting times at health facilities, poor attitudes of health workers, the perception that care was unnecessary due to the absence of complications, and associated costs (e.g., transport)** (moderate confidence) [34,52,55–58,62,65,69,74,75,78,85,86,89,93].

**Prevention and/or treatment of anaemia in pregnant women through diet, supplementation, or clinical intervention**

Findings 2.1 to 2.15 are categorised under this theme, with 58 contributing studies. These studies explore key stakeholders' perceptions and experiences of the prevention and treatment of anaemia in pregnant women through diet, supplements (primarily oral iron and folic acid, as well as oral iron and multiple micronutrient supplements), and clinical interventions (blood transfusion and intravenous iron).

*Finding 2.1: Importance of a nutritious diet to manage anaemia in pregnancy.* **Many health workers, lay health workers, women, and community members described that a nutritious diet was essential for preventing and/or treating anaemia in pregnant women, as they shared their food and blood with the fetus. Some women's diets changed during pregnancy to include more vegetables and fruits, larger portions, and to eat first rather than last within their families. Some women expressed a preference for managing anaemia through diet rather than supplementation, viewing food as natural and containing all necessary iron** (moderate confidence) [34,36,43,44,46,52,56–58,66,70,73–75,78,79,81,85,87,88,90,91,93,94].

Some health workers and community members were concerned that a lack of a nutritious diet during pregnancy could lead to long-term complications for children [56].

*Finding 2.2: Challenges to women having a nutritious diet to manage anaemia in pregnancy.* **Many women and some community members, health workers, and lay health workers described how the high cost of nutritious food was a barrier to women having a nutrient-rich diet to prevent and/or treat anaemia in pregnancy.**

**Additional barriers to women having a nutritious diet in pregnancy included limited decision-making power and social positionality of women, low availability of nutritious foods in some settings, inadequate knowledge of iron-rich foods, consumption of food and drink that inhibited iron absorption, and food taboos and religious beliefs that prevented pregnant woman from eating certain foods (e.g., animal products)** (moderate confidence) [35,42,44,46,50–54,56,57,65,70,73–75,77,81,82,85,88,93,94].

Other beliefs that limited women from having a nutritious diet included the idea that weight gain, rather than consuming iron-rich foods, was the best way to manage anaemia during pregnancy [54]. Conversely, some women and community members believed that overeating during pregnancy could reduce space for the fetus, increasing the likelihood of requiring a caesarean section [52].

*Finding 2.3: Characteristics of supplements.* **Some women discussed the bitter taste, smell, and large size of supplements as negatively affecting adherence** (low confidence) [35,38,39,42,43,47,52,59,60,63,75].

*Finding 2.4: Side-effects of supplements.* **Many women reported not adhering to supplements after experiencing side-effects, including nausea, dizziness, vomiting, diarrhoea, constipation, blackened stool, heartburn, loss or increase of appetite, and/or indigestion** (moderate confidence) [34–39,41–43,45–47,52,53,57–60,62,63,72–75,84–86,88,93].

When women experienced side-effects from supplements, they would either skip them or completely discontinue use. Some women reported that side-effects from supplements subsided over time [47,52].

*Finding 2.5: Misconceptions about supplements.* **Women, community members, and some health workers and lay health workers had misconceptions about supplements that limited women from taking them to prevent and/ or treat anaemia in pregnancy. Common misconceptions included that supplements would increase the size of the fetus and result in women having a difficult childbirth, were only needed to treat women with anaemia, could cause miscarriage, were bad for the fetus or woman's health, or increased maternal blood and therefore bleeding during birth. These misconceptions were often told to women by family members (e.g., mother-in-law)** (moderate confidence) [35,37,43,46,47,50,52,57,59,60,62,63,65,85,86,90,92–94].

Health workers and lay health workers reported not providing supplements [iron and folic acid] to women who were not anaemic, which indirectly reinforced the misconception that supplements were only for treating anaemia [47,60]. Additionally, some health workers prohibited women from taking supplements [iron and folic acid] in the first trimester for fear of being blamed if a miscarriage occurred [85].

*Finding 2.6: Challenges to women remembering to take supplements.* **Many women forgot to take supplements consistently, often due to competing daily activities, such as household chores and work, caring for children, being away from home, or taking other medications** (moderate confidence) [36,38,39,41,43,47,53,54,57,59,62,63,68,72–75,84,86,92,93].

*Finding 2.7: Inadequate counselling from health workers and lay health workers about supplements.* **Women received insufficient and/or inconsistent information on – why they were provided with the supplements, the benefits of taking supplements, when they should start taking supplements, how often they should take supplements, potential side-effects of supplements, and how side-effects were minimised** (moderate confidence) [35,38,43,47,54,57,58,60,62,73–75,86,89,90,93].

Women described receiving minimal information about supplements, such as that iron and folic acid were *"good"* for the mother and fetus and that they *"boost the blood"* [73,75].

*Finding 2.8: Insufficient supply of supplements.* **Many health workers, lay health workers, and some women described an insufficient supply of supplements, which resulted in health workers and lay health workers prioritising the distribution of supplements to anaemic pregnant women over non-anaemic pregnant women, providing women with an inadequate quantity of supplements, or referring women to other facilities to purchase supplements out-of-pocket (e.g., private drug stores)** (moderate confidence) [35,41,45,46,49,55,59,60,62,64,69,71,73,75,77,85,86,91,93].

Some health workers and lay health workers outlined reasons why there was an insufficient supply of supplements, including issues with forecasting, procurement, and storage [45,46,49,55,60,69,71].

*Finding 2.9: Women's knowledge of supplements and experience of positive benefits.* **Adherence to supplements was motivated by women's knowledge of their health benefits, particularly for the fetus, and for the prevention of illness and excessive blood loss during childbirth. Some women discussed how the relief of anaemia symptoms after taking supplements (e.g., no longer feeling weak or lightheaded) improved adherence** (moderate confidence) [36,38,41,47,52,53,60,62,68,73–75,85,86,88,90–94].

A woman described *"having a lot of energy"* and *"no longer sleeping during the day" s*ince starting supplements [multiple micronutrient supplements] [47].

*Finding 2.10: Influence of family support on women taking supplements.* **Many women, health workers, lay health workers, and community members highlighted how women's adherence to supplements was influenced by whether they had received encouragement and/or reminders to take supplements from family members (most commonly their husband, mother-in-law, or parents)** (moderate confidence) [38,39,43,47,50–52,57,63,66,68,73,74,76,82,86,92,94].

Women described how encouragement from family members to take supplements [multiple micronutrient supplements] showed that they cared, which motivated adherence [47]. Conversely, women outlined that family members could prohibit them from taking supplements [iron and folic acid] [43,57,92].

*Finding 2.11: Reminder strategies to take supplements.* **Some women and health workers perceived reminder strategies as facilitating women's adherence to supplements. Reminder strategies included linking taking supplements with other daily behaviour (e.g., with mealtime) and storing supplements in a visible location** (very low confidence) [36,57,73,75].

*Finding 2.12: Recommendation from health workers to take supplements.* **Many women reported taking supplements because they trusted and accepted the advice of health workers** (moderate confidence) [38,41,52,57,59,60,62,63,68,73,74,85–88,94].

Women described taking supplements because they trusted health workers, especially doctors, who were perceived as highly educated and knowing what was best for them [41,57,59,60,63,88].

*Finding 2.13: Alternative distribution methods for supplements.* **Most women, health workers, and lay health workers supported distributing supplements to women during home visits. Lay health workers discussed how women's adherence to supplements improved with home visits, as they also allowed for health education and counselling. Some women discussed how community-based distribution mitigated barriers to receiving supplements, such as distance to health facilities and transportation costs** (very low confidence) [47,64,73,89].

A challenge for lay health workers distributing supplements to women during home visits was the lack of remuneration [64]. Conversely, women described how home visits saved them time and money compared to visiting a health facility [47].

*Finding 2.14: Blood transfusion as treatment for severe anaemia in pregnant women.* **Women and health workers perceived blood transfusion as treatment for women with severe anaemia; however, cost, religion, blood and donor availability, fear of contracting diseases, and a woman's social position limited access** (low confidence) [34,39,45,52,54,57,80,87,93].

Some women perceived a blood transfusion as the last treatment option for severe anaemia: *"If blood has finished and you can't buy it, you simply die"* [87].

*Finding 2.15: Intravenous iron as an alternative treatment option for iron-deficiency anaemia in pregnant women.* **Health workers, women, and community members described intravenous iron as a treatment option for women who had iron-deficiency anaemia and were unable to take supplements (e.g., due to side-effects), who attended antenatal care late, or who were unable to have a blood transfusion because of religious beliefs** (low confidence) [39,40,48,56,57,80,90].

The perceived advantages of intravenous iron over other anaemia treatment options (e.g., oral iron, blood transfusion) included better adherence and immediate improvements in health for pregnant women [39,48]. However, disadvantages included misconceptions (e.g., could cause a miscarriage), the need for adequate human and physical resources, and the potential risk of serious adverse events [39,48]. Factors influencing health workers' implementation fidelity of intravenous iron included teamwork, availability of charts and protocols, space for storing resuscitation materials, high workload, and prolonged waiting times for pregnant women [40]. Suggested strategies for implementing intravenous iron treatment included community sensitisation, peer advocacy, male (husband) involvement, creating an accessible package of required equipment and medicines, and specialised training for health workers [39,48].

**Testing pregnant women for anaemia**

Findings 3.1 and 3.2 are categorised under this theme, with 11 studies exploring the benefits and challenges of using point-of-care devices used with capillary blood and full blood count using venous blood to test pregnant women for anaemia.

*Finding 3.1: Benefits of testing pregnant women for anaemia with point-of-care devices using capillary blood.* **Health workers provided with anaemia point-of-care devices (portable haemoglobinometers or a colour scale) felt they were easy to use with capillary blood and were eager to meet pregnant women's antenatal care needs. Health workers described how more women were diagnosed with anaemia as they were no longer lost to follow-up during the referral process to laboratories for a full blood count using venous blood. Women described how haemoglobin results from a point-of-care test improved their trust in the anaemia diagnosis and increased their involvement in the treatment decision-making process** (very low confidence) [61,69].

Health workers described how they were able to easily integrate point-of-care testing using capillary blood into their workflow [61,69].

*Finding 3.2: Challenges to testing pregnant women for anaemia with point-of-care devices using capillary blood and full blood count using venous blood.* **Challenges to health workers using anaemia point-of-care devices with capillary blood included a shortage of tests, staff, and equipment (e.g., cuvettes required for most haemoglobinometers), inadequate training, and heavy workloads. For full blood count using venous blood, challenges described by health workers included a lack of laboratory facilities and women not attending facilities when referred. Health workers expressed reservations about the accuracy of results for both types of anaemia tests** (low confidence) [34,45,46,61,65,67,69,75,77,82,83].

For point-of-care devices, there were reservations about the quality and accuracy of results [61,67]. For full blood count, there were concerns that human error (e.g., mixing of samples) and technical problems with the machine could lead to an inaccurate result [34,61]. Challenges experienced by health workers led some to use clinical assessment (inspection of the conjunctiva, tongue, and palms) to diagnose anaemia in pregnant women [34,45,61,67]. Health workers described clinical assessment as *"quicker and cheaper"* than other testing methods [61]. However, there were doubts about its subjectivity and its ability to diagnose anaemia other than in severe cases [34,45].

**Factors affecting health workers' engagement in the management of anaemia in pregnant women**

Findings 4.1 and 4.2 are categorised under this theme, with 20 contributing studies exploring the training and guidance health workers receive and the challenges they experience in managing anaemia in pregnant women.

*Finding 4.1: Training and guidance on how to manage anaemia in pregnant women among health workers and lay health workers.* **Some health workers and lay health workers described receiving inadequate training on how to manage anaemia in pregnant women (e.g., on counselling, supplements, and/or testing). They also described a lack of specific or standardised guidelines on how to manage anaemia in pregnant women** (moderate confidence) [34,45,49,61,62,64,65,67,69,71,73,75,77,80,82,85].

Health workers and lay health workers who had recently received adequate training reported greater confidence in counselling pregnant women about anaemia and had adjusted their management practices (e.g., supplying women with the correct number of supplements or administering intravenous iron) [49,80].

Finding 4.2: Health worker and lay health worker challenges to managing anaemia in pregnant women. **Many health workers and lay health workers described how staffing shortages, heavy workloads, insufficient facilities and space, and poor work conditions affected their ability to manage anaemia in pregnant women** (moderate confidence) [34,45,55,57,60,62,65,67,69,71,73,77,82,83].

For example, a health worker described how no one was assigned to manage anaemia in pregnant women or to promote community awareness, unlike for HIV and malaria [62].

## Discussion

Our review presents a comprehensive exploration of women's, health workers', and other key stakeholders' perceptions of anaemia in pregnant women, as well as experiences of preventing, testing, or treating anaemia in pregnant women. We identified challenges to health workers managing anaemia in pregnant women, including staffing shortages and inadequate training. Health workers also described difficulties in testing pregnant women for anaemia (point-of-care devices and full blood count), due to shortages of tests and equipment and reservations about the accuracy of results. We found that many women perceived a nutritious diet as integral to preventing and/or treating anaemia; however, high cost and food taboos were often barriers. Women outlined reasons for not adhering to supplements (most commonly iron and folic acid), including side-effects and difficulty in remembering to take them. Factors that improved women's adherence to supplements included family support and recommendations from health workers. We found that blood transfusion was perceived as treatment for women with severe anaemia whilst intravenous iron was viewed as a treatment option for women who had iron-deficiency anaemia and were unable to take supplements, attended antenatal care late, or could not receive a blood transfusion. Lastly, our review showed the influence of the socio-cultural context on the management of anaemia in pregnancy, particularly the limited decision making power and social position of women and delayed or inconsistent antenatal care.

Our finding that the socio-cultural context influences the management of anaemia in pregnant women is aligned with Downe and colleagues' [95] research, which found that traditional beliefs and practices, gender issues, and selective use of antenatal care affected the quality of care women received during pregnancy [95]. In the context of iron-deficiency anaemia, delayed or inconsistent antenatal care is particularly problematic, as supplements can require up to six months to replenish iron stores [96]. Further, Downe and colleagues [95] found that across diverse settings and contexts, women viewed pregnancy as a normal and healthy state of being. This perspective may explain our finding of women normalising symptoms of anaemia and seeking treatment only when symptoms were severe. Overall, our findings highlight that the management of anaemia is deeply shaped by the broader context in which pregnancy occurs and demonstrates the importance of interventions that address non-biomedical factors [97].

An estimated 50–70% of pregnant women do not adhere to taking daily oral iron supplements [6,18]. Our findings provide context to these suboptimal adherence rates, with women describing experiences of side-effects and difficulties in remembering to take supplements. Consistent with a review on calcium supplementation during pregnancy [98], we identified that knowledge of supplements and experience of positive benefits, family support, and reminder strategies improved adherence. Overall, our study contributes to the existing body of evidence on factors affecting pregnant women's engagement with supplementation, complementing a previous study that reviewed barriers and enablers of oral iron and folic acid use across seven countries in Africa and Asia [19].

Compared to supplements, we found that intravenous iron was perceived to have some advantages as a treatment option for iron-deficiency anaemia, including better adherence and immediate improvements in health for pregnant women. Unlike supplements, intravenous iron does not cause gastrointestinal side-effects, and some modern formulations

can be administered in a single dose, rapidly repleting iron stores and haemoglobin levels [99–101]. However, intravenous iron as a treatment option for iron-deficiency anaemia is typically not available in LMICs, with clinical trials, implementation science studies, and cost-effectiveness analyses only recently being conducted in Malawi, Nigeria, Bangladesh and India [102–109].

We found that health workers often worked in the context of structural health-system deficiencies, including staffing shortages and inadequate training, which limited their ability to manage anaemia in pregnant women. Human and physical resource shortages are consistent challenges to health workers delivering evidence-based interventions and quality care in low-resource settings [95,110,111]. For example, we found that some women did not receive supplements, and even when they did, they often did not receive adequate counselling from health workers on how to take them. Inadequate training and the absence of anaemia management guidelines may also explain why some health workers had misconceptions about the accuracy of full blood count using venous blood (WHO recommended testing method), preferring instead to use clinical assessment (not a WHO recommended testing method) [5].

## Implications for research and policy

Future research could map our findings to implementation science frameworks and models, such as the Theoretical Domains Framework and the Capability, Opportunity, and Motivation of Behaviour model [112,113]. This approach would help identify behavioural determinants related to managing anaemia in pregnant women. The Behaviour Change Wheel, an intervention development tool, could then be used to select appropriate intervention types – such as training or environmental restructuring – that specifically targets these determinants and inform the design of potential implementation strategies [113].

Qualitative and mixed-methods research could explore whether factors influencing women's adherence to supplements vary by type (e.g., iron and folic acid versus multiple micronutrient supplements) and dosage (e.g., daily versus intermittent). Our review has also revealed limited evidence on stakeholders' perceptions and experiences of anaemia testing (e.g., challenges associated with point-of-care tests in determining the underlying cause of anaemia) and clinical interventions, specifically blood transfusion and intravenous iron. Further research, particularly from the perspective of pregnant women, is needed to understand how anaemia testing and clinical interventions are currently functioning and how they can be improved.

Our findings could contribute to the development of future WHO recommendations, particularly around the domains of acceptability, feasibility, values, and implementation considerations [20]. For example, findings 2.3 to 2.13 may inform the assessment of the acceptability of iron and folic acid supplements for the prevention and treatment of anaemia, whereas findings 2.15, 4.1, and 4.2 might support considerations of the feasibility of intravenous iron treatment.

## Strengths and limitations

Our search strategies resulted in extensive coverage of published qualitative literature, with no restrictions on language, geographical setting, or publication date. We therefore did not conduct a grey literature search, which could have further broadened our evidence. Most of the included studies were conducted in countries in Africa and South-East Asia, the regions with the highest burden of anaemia [4]. Only a few included studies were conducted in countries in the Americas, Eastern Mediterranean, Western Pacific, and Europe. A key strength of our review is the use of the GRADE-CERQual approach to assess our confidence in each review finding [27,28]. The rigour and proven utility of this approach increases the likelihood that our findings will inform research and policy.

## Conclusion

Our review comprehensively explores the perceptions and experiences of women, health workers, and other key stakeholders on the management of anaemia in pregnant women. Findings may contribute to the development of future WHO



recommendations, particularly around the domains of acceptability, feasibility, values, and implementation considerations. Future research could use the review findings alongside implementation science frameworks to develop effective strategies for improving the prevention, testing, and/or treatment of anaemia in pregnant women.

## Supporting information

**S1 Appendix.  Preferred reporting items for systematic reviews and meta-analyses (PRISMA) reporting checklist.**
(DOCX)

**S2 Appendix.  Search strategies.**
(DOCX)

**S3 Appendix.  GRADE-CERQual evidence profile.**
(DOCX)

**S4 Appendix.  Characteristics of included papers.**
(DOCX)

**S5 Appendix.  Methodological limitations of included studies.**
(DOCX)

## Acknowledgments

We thank Lindy Cochrane (Information Specialist) for her assistance with developing the search criteria. We thank Nicole Minckas for verifying the translation accuracy of the Spanish study.

## Author contributions

**Conceptualization:** Ebony Verbunt, Özge Tunçalp, Lisa M. Rogers, Meghan A. Bohren.

**Data curation:** Ebony Verbunt, Martha Vazquez Corona, Meghan A. Bohren.

**Formal analysis:** Ebony Verbunt, Martha Vazquez Corona, Meghan A. Bohren.

**Methodology:** Ebony Verbunt, Meghan A. Bohren.

**Project administration:** Ebony Verbunt.

**Supervision:** Meghan A Bohren.

**Validation:** Ebony Verbunt, Martha Vazquez Corona, Meghan A. Bohren.

**Visualization:** Ebony Verbunt.

**Writing – original draft:** Ebony Verbunt.

**Writing – review & editing:** Ebony Verbunt, Martha Vazquez Corona, Özge Tunçalp, Lisa M. Rogers, Khic-Houy Prang, Cathy Vaughan, Meghan A. Bohren.

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
