## [Decision Letter · Decision Letter 0]

15 Apr 2025

PGPH-D-25-00238

Perceptions and experiences of the prevention, testing, and treatment of anaemia in pregnant women: a qualitative evidence synthesis

Dear Dr. Ebony Verbunt,

Thank you for submitting your manuscript to PLOS Global Public Health. After careful consideration, we feel that it has merit but does not fully meet PLOS Global Public Health’s publication criteria as it currently stands. Therefore, we invite you to submit a revised version of the manuscript that addresses the points raised during the review process.

We look forward to receiving your revised manuscript.

Kind regards,

Aditi Apte, MD PhD

Academic Editor

Journal Requirements:

1. Please identify your study as "systematic review" in the title of your manuscript.

Additional Editor Comments (if provided):

Reviewers' comments:

Reviewer's Responses to Questions

**Comments to the Author**

1. Does this manuscript meet PLOS Global Public Health’s publication criteria ? Is the manuscript technically sound, and do the data support the conclusions? The manuscript must describe methodologically and ethically rigorous research with conclusions that are appropriately drawn based on the data presented.

Reviewer #1: Yes

Reviewer #2: Yes

2. Has the statistical analysis been performed appropriately and rigorously?

Reviewer #1: Yes

Reviewer #2: N/A

3. Have the authors made all data underlying the findings in their manuscript fully available (please refer to the Data Availability Statement at the start of the manuscript PDF file)?

Reviewer #1: Yes

Reviewer #2: Yes

4. Is the manuscript presented in an intelligible fashion and written in standard English?

Reviewer #1: Yes

Reviewer #2: Yes

5. Review Comments to the Author

Reviewer #1: I would like to congratulate the authors for this important work. The findings are interesting and represent the issues commonly faced on ground. I have only one small recommendation: A significant barrier to anemia management is the lack of a POCT that could screen for the cause of anemia. Available POCT (digital hemoglobinometer) in LMICs only screens for the presence of anemia following which the mother is given iron therapy. Available, evidence from India suggests that the prevalence of iron deficiency in mothers with anemia might only be 10-40%. This means that a significant proportion of mothers are getting iron unnecessarily which adds to the GI side effects and also the has the potential to alter gut flora (something that has not been studied much). The existing guidelines in India are also out dated and recommend twice daily iron in anemic mothers when there is enough evidence to show that alternate day iron might work better. I would suggest the authors to include this in the discussion of the paper and recommend future studies on anemia POCTs and oral therapy of iron before focusing on IV iron therapy as there is heavy pharma industry influence in IV iron therapy which in my opinion will fail to improve anemia in LMICs and only add to the cost of anemia management for reasons cited above - only 10-40% have IDA and oral therapy improvement has not been studied extensively.

Besides this, I have no other comments. Thank you for the opportunity to review this paper.

Reviewer #2: The ongoing prevalence of anemia during pregnancy, especially in low- and middle-income nations, is a significant worldwide public health concern that is addressed in this publication. The qualitative evidence synthesis has the potential to shape WHO guidelines and prenatal care programs because it is methodologically sound, well-written, and instructive. To improve receptivity and interpretability, a few points need to be clarified,  and slightly revised.

1. To support your case of added value, clearly compare your findings with preceding research (such as earlier thematic reviews or individual studies) in the Discussion.

2. The section on reflexivity is good. Could be improved by carefully considering how the study participants' professional or cultural proximity may have influenced how the data was interpreted. It reads as general at the moment.

3. More specific, feasible recommendations would strengthen the conclusion. What particular interventions (such as task-shifting or mHealth reminders) might be tested in light of the results, for example?

4. Table 2 is arranged neatly. But some conclusions (such 2.11, 2.13, and 3.1) are based on limited evidence. Think about using a footnote or asterisk to visually underline the constraint.

This is a valuable contribution to maternal health policy discourse and qualitative synthesis methods in global health.

6. PLOS authors have the option to publish the peer review history of their article (what does this mean? ). If published, this will include your full peer review and any attached files.

**Do you want your identity to be public for this peer review?** For information about this choice, including consent withdrawal, please see our Privacy Policy .

Reviewer #1: **Yes: ** Parth Sharma

Reviewer #2: No

---

## [Decision Letter · Decision Letter 1]

25 Jun 2025

PGPH-D-25-00238R1

Perceptions and experiences of the prevention, testing, and treatment of anaemia in pregnant women: a qualitative evidence synthesis

Dear Dr. Verbunt,

Thank you for submitting your manuscript to PLOS Global Public Health. After careful consideration, we feel that it has merit but does not fully meet PLOS Global Public Health’s publication criteria as it currently stands. Therefore, we invite you to submit a revised version of the manuscript that addresses the points raised during the review process.

Reviewer 2 has recommended a few additional revisions to improve the clarity and conciseness of your manuscript, and also suggests including more clear policy recommendations or priority actions specific to the WHO guideline component of the health system.

We look forward to receiving your revised manuscript.

Kind regards,

Jennifer Tucker, PhD

Staff Editor

Journal Requirements:

1. Please identify your study as "systematic review" in the title of your manuscript.

Additional Editor Comments (if provided):

Reviewers' comments:

Reviewer's Responses to Questions

**Comments to the Author**

1. If the authors have adequately addressed your comments raised in a previous round of review and you feel that this manuscript is now acceptable for publication, you may indicate that here to bypass the “Comments to the Author” section, enter your conflict of interest statement in the “Confidential to Editor” section, and submit your "Accept" recommendation.

Reviewer #2: All comments have been addressed

2. Does this manuscript meet PLOS Global Public Health’s publication criteria ? Is the manuscript technically sound, and do the data support the conclusions? The manuscript must describe methodologically and ethically rigorous research with conclusions that are appropriately drawn based on the data presented.

Reviewer #2: Yes

3. Has the statistical analysis been performed appropriately and rigorously?

Reviewer #2: Yes

4. Have the authors made all data underlying the findings in their manuscript fully available (please refer to the Data Availability Statement at the start of the manuscript PDF file)?

Reviewer #2: Yes

5. Is the manuscript presented in an intelligible fashion and written in standard English?

Reviewer #2: No

6. Review Comments to the Author

Reviewer #2: The abstracts and parts of the main document are too lengthy and can be improved with more concise and focused writing.

Include more concrete policy recommendations or priority actions specific to the WHO guideline component of the health system.

7. PLOS authors have the option to publish the peer review history of their article (what does this mean? ). If published, this will include your full peer review and any attached files.

**Do you want your identity to be public for this peer review?** For information about this choice, including consent withdrawal, please see our Privacy Policy .

Reviewer #2: No

---

## [Decision Letter · Decision Letter 2]

7 Aug 2025

PGPH-D-25-00238R2

Perceptions and experiences of the prevention, testing, and treatment of anaemia in pregnant women: a qualitative evidence synthesis

Dear Dr. Verbunt,

Thank you for submitting your manuscript to PLOS Global Public Health. After careful consideration, we feel that it has merit but does not fully meet PLOS Global Public Health’s publication criteria as it currently stands. Therefore, we invite you to submit a revised version of the manuscript that addresses the points raised during the review process.

We look forward to receiving your revised manuscript.

Kind regards,

Helen Howard

Staff Editor

Journal Requirements:

Additional Editor Comments (if provided):

Reviewers' comments:

Reviewer's Responses to Questions

**Comments to the Author**

1. If the authors have adequately addressed your comments raised in a previous round of review and you feel that this manuscript is now acceptable for publication, you may indicate that here to bypass the “Comments to the Author” section, enter your conflict of interest statement in the “Confidential to Editor” section, and submit your "Accept" recommendation.

Reviewer #3: All comments have been addressed

2. Does this manuscript meet PLOS Global Public Health’s publication criteria ? Is the manuscript technically sound, and do the data support the conclusions? The manuscript must describe methodologically and ethically rigorous research with conclusions that are appropriately drawn based on the data presented.

Reviewer #3: Yes

3. Has the statistical analysis been performed appropriately and rigorously?

Reviewer #3: N/A

4. Have the authors made all data underlying the findings in their manuscript fully available (please refer to the Data Availability Statement at the start of the manuscript PDF file)?

Reviewer #3: Yes

5. Is the manuscript presented in an intelligible fashion and written in standard English?

Reviewer #3: Yes

6. Review Comments to the Author

Reviewer #3: This is an excellent synthesis of the available qualitative data on iron interventions for the prevention and treatment of anaemia in pregnancy.

Can the authors provide a bit more detail in the discussion about the wider context of their work. This could include the types of interventions, their availability and high level summaries of the use case for these? For an audience who arent aware of the nuances in this area of research this would be helpful.

Can the authors also provide geographic context for the included studies? I think this is mentioned in the narrative in the section on description of studies, but it would also be helpful to understand if the themes and review findings were across all regions, or in specific areas.

There are some areas of repetition (the data described in table 2 are repeated in the narrative). It would be helpful if other information is provided to make clear if such repetition is needed, or actually if additional details are provided that would be preferrable. The authors could consider condensing, however will need to make a judgement if important details are then lost.

The discussion is short, and provides little context, so could be expanded for the non expert reader, word count permitting.

7. PLOS authors have the option to publish the peer review history of their article (what does this mean? ). If published, this will include your full peer review and any attached files.

**Do you want your identity to be public for this peer review?** For information about this choice, including consent withdrawal, please see our Privacy Policy .

Reviewer #3: **Yes: ** Jahnavi Daru

---

## [Editor Report · Decision Letter 3]

19 Aug 2025

Perceptions and experiences of the prevention, testing, and treatment of anaemia in pregnant women: a qualitative evidence synthesis

PGPH-D-25-00238R3

Dear Ms Verbunt,

We are pleased to inform you that your manuscript 'Perceptions and experiences of the prevention, testing, and treatment of anaemia in pregnant women: a qualitative evidence synthesis' has been provisionally accepted for publication in PLOS Global Public Health.

Best regards,

Julia Robinson

Executive Editor